# A SIMPLE APPROACH TO DEFINE CURRICULA FOR TRAINING NEURAL NETWORKS

## ABSTRACT

In practice, sequence of mini-batches generated by uniform sampling of examples from the entire data is used for training neural networks. Curriculum learning is a training strategy that sorts the training examples by their difficulty and gradually exposes them to the learner. In this work, we propose two novel curriculum learning algorithms and empirically show their improvements in performance with convolutional and fully-connected neural networks on multiple real image datasets. Our dynamic curriculum learning algorithm tries to reduce the distance between the network weight and an optimal weight at any training step by greedily sampling examples with gradients that are directed towards the optimal weight. The curriculum ordering determined by our dynamic algorithm achieves a training speedup of $\sim 45\%$ in our experiments. We also introduce a new task-specific curriculum learning strategy that uses statistical measures such as standard deviation and entropy values to score the difficulty of data points in natural image datasets. We show that this new approach yields a mean training speedup of $\sim 43\%$ in the experiments we perform. Further, we also use our algorithms to learn why curriculum learning works. Based on our study, we argue that curriculum learning removes noisy examples from the initial phases of training, and gradually exposes them to the learner acting like a regularizer that helps in improving the generalization ability of the learner.

## 1 INTRODUCTION

Stochastic Gradient Descent (SGD) (Robbins & Monro, 1951) is a simple yet widely used algorithm for machine learning optimization. There have been many efforts to improve its performance. A number of such directions, such as AdaGrad (Duchi et al., 2011), RMSProp (Tieleman & Hinton, 2012), and Adam (Kingma & Ba, 2014), improve upon SGD by fine-tuning its learning rate, often adaptively. However, Wilson et al. (2017) has shown that the solutions found by adaptive methods generalize worse even for simple overparameterized problems. Reddi et al. (2019) introduced AMSGrad hoping to solve this issue. Yet there is performance gap between AMSGrad and SGD in terms of the ability to generalize (Keskar & Socher, 2017). Further, Choi et al. (2019) shows that more general optimizers such as Adam and RMSProp can never underperform SGD when all their hyperparameters are carefully tuned. Hence, SGD still remains one of the main workhorses of the ML optimization toolkit.

SGD proceeds by stochastically making unbiased estimates of the gradient on the full data (Zhao & Zhang, 2015). However, this approach does not match the way humans typically learn various tasks. We learn a concept faster if we are presented the easy examples first and then gradually exposed to examples with more complexity, based on a curriculum. An orthogonal extension to SGD (Weinshall & Cohen, 2018), that has some promise in improving its performance is to choose examples according to a specific strategy, driven by cognitive science – this is curriculum learning (CL) (Bengio et al., 2009), wherein the examples are shown to the learner based on a curriculum.

### 1.1 RELATED WORKS

Bengio et al. (2009) formalizes the idea of CL in machine learning framework where the examples are fed to the learner in an order based on its *difficulty*. The notation of difficulty of examples

has not really been formalized and various heuristics have been tried out: Bengio et al. (2009) uses manually crafted scores, self-paced learning (SPL) (Kumar et al., 2010) uses the loss values with respect to the learner's current parameters, and CL by transfer learning uses the loss values with respect to a pre-trained learner to rate the difficulty of examples in data. Among these works, what makes SPL particular is that they use a dynamic CL strategy, i.e. the preferred ordering is determined dynamically over the course of the optimization. However, SPL does not really improve the performance of deep learning models, as noted in (Fan et al., 2018). Similarly, Loshchilov & Hutter (2015) uses a function of rank based on latest loss values for online batch selection for faster training of neural networks. Katharopoulos & Fleuret (2018) and Chang et al. (2017) perform importance sampling to reduce the variance of stochastic gradients during training. Graves et al. (2017) and Matiisen et al. (2020) propose teacher-guided automatic CL algorithms that employ various supervised measures to define dynamic curricula. The most recent works in CL show its advantages in reinforcement learning (Portelas et al., 2020; Zhang et al., 2020).

The recent work by Weinshall & Cohen (2018) introduces the notion of *ideal difficult score* to rate the difficulty of examples based on the loss values with respect to the set of optimal hypotheses. They theoretically show that for linear regression, the expected rate of convergence at a training step $t$ for an example monotonically decreases with its ideal difficulty score. This is practically validated by Hacohen & Weinshall (2019) by sorting the training examples based on the performance of a network trained through transfer learning. However, there is a lack of theory to show that CL improves the performance of a completely trained network. Thus, while CL indicates that it is possible to improve the performance of SGD by a judicious ordering, both the theoretical insights as well as concrete empirical guidelines to create this ordering remain unclear.

While the previous CL works employ tedious methods to score the difficulty level of the examples, Hu et al. (2020) uses the number of audio sources to determine the difficulty for audiovisual learning. Liu et al. (2020) uses the norm of word embeddings as a difficulty measure for CL for neural machine translation. In light of these recent works, we discuss the idea of using task-specific statistical (unsupervised) measures to score examples making it easy to perform CL on real image datasets without the aid of any pre-trained network.

## 1.2 OUR CONTRIBUTIONS

Our work proposes two novel algorithms for CL. We do a thorough empirical study of our algorithms and provide some more insights into why CL works. Our contributions are as follows:

- We propose a novel **dynamic curriculum learning** (DCL) algorithm to study the behaviour of CL. DCL is not a practical CL algorithm since it requires the knowledge of a reasonable local optima as needs to compute the gradients of full data after ever training epoch. DCL uses the gradient information to define a curriculum that minimizes the distance between the current weight and a desired local minima. However, this simplicity in the definition of DCL makes it easier to analyze its performance formally.

- Our DCL algorithm generates a natural ordering for training the examples. Previous CL works have demonstrated that exposing a part of the data initially and then gradually exposing the rest is a standard way to setup a curriculum. We use two variants of our DCL framework to show that it is not just the subset of data which is exposed to the model that matters, but also **the ordering within the data partition that is exposed**. We also analyze how DCL is able to serve as a regularizer and **improve the generalization of networks**.

- We contribute a simple, novel and practical CL approach for image classification tasks that does the **ordering of examples in a completely unsupervised manner using statistical measures**. Our insight is that statistical measures could have an association with the difficulty of examples in real data. We empirically analyze our argument of using statistical scoring measures (especially standard deviation) over permutations of multiple datasets and networks. Additionally, we study why CL based on standard deviation scoring works using our DCL framework.

**Algorithm 1** Approximate greedy dynamic curriculum learning (*DCL+*).

**Input:** Data $\mathcal{X}$, local minima $\tilde{w}$, weight $w_t$, batch size $b$, and pacing function *pace*.

**Output:** Sequence of mini-batches $B_t$ for the next training epoch.

1: $\tilde{\boldsymbol{a}}_t \leftarrow \tilde{\boldsymbol{w}} - \boldsymbol{w}_t$
2: $\rho_t \leftarrow [\,]$
3: $B_t \leftarrow [\,]$
4: **for** $(i = 0;\ N;\ 1)$ **do**
5:     append $-\dfrac{\tilde{\boldsymbol{a}}_t^{\mathrm{T}} \cdot \nabla f_i(\boldsymbol{w}_t)}{\|\tilde{\boldsymbol{a}}_t\|_2}$ to $\rho_t$
6: **end for**
7: $\tilde{\mathcal{X}} \leftarrow \mathcal{X}$ sorted according to $\rho_t$, in ascending order
8: $size \leftarrow pace(t)$
9: **for** $(i = 0;\ size;\ b)$ **do**
10:     append $\tilde{\mathcal{X}}[i, ..., i+b-1]$ to $B_t$
11: **end for**
12: **return** $B_t$

Figure 1: A geometrical interpretation of gradient steps for the understanding of equation 1.

## 2 PRELIMINARIES

At any training step $t$, SGD updates the weight $\boldsymbol{w}_t$ using $\nabla f_i(\boldsymbol{w}_t)$ which is the gradient of loss of example $\boldsymbol{x}_i$ with respect to the current weight. The learning rate and the data are denoted by $\eta$ and $\mathcal{X} = \{(\boldsymbol{x}_i, y_i)\}_{i=0}^{N-1}$ respectively, where $\boldsymbol{x}_i \in \mathbb{R}^d$ denotes a data point and $y_i \in [K]$ its corresponding label for a dataset with $K$ classes. We denote the learner as $h_\vartheta : \mathbb{R}^d \rightarrow [K]$. Generally, SGD is used to train $h_\vartheta$ by giving the model a sequence of mini-batches $\{B_0, B_1, ..., B_{T-1}\}$, where $B_i \subseteq \mathcal{X}\ \forall i \in [T]$. Each $B_i$ is generated by uniformly sampling examples from the data. We denote this approach as *vanilla*.

In CL, the curriculum is defined by two functions, namely the scoring function and the pacing function. The scoring function, $score_\vartheta(\boldsymbol{x}_i, y_i) : \mathbb{R}^d \times [K] \rightarrow \mathbb{R}$, scores each example in the dataset. Scoring function is used to sort $\mathcal{X}$ in an ascending order of difficulty. A data point $(\boldsymbol{x}_i, y_i)$ is said to be easier than $(\boldsymbol{x}_j, y_j)$ if $score_\vartheta(\boldsymbol{x}_i, y_i) < score_\vartheta(\boldsymbol{x}_j, y_j)$, where both the examples belong to $\mathcal{X}$. Unsupervised scoring measures do not use the data labels to determine the difficulty of data points. The pacing function, $pace_\vartheta(t) : [T] \rightarrow [N]$, determines how much of the data is to be exposed at a training step $t \in [T]$.

We define speedup for CL model as its improvement over *vanilla* model (in terms of the number of training steps) to achieve a given test accuracy. For example, CL has $2\times$ speedup if *vanilla* model achieves 90% test accuracy in 100 training steps while CL achieves the same 90% test accuracy in 50 training steps.

## 3 DYNAMIC CURRICULUM LEARNING

For DCL algorithms (Kumar et al., 2010; Graves et al., 2017; Matiisen et al., 2020), examples are scored and sorted after every few training steps since the parameters of the scoring function change dynamically with the learner as training proceeds. Hacohen & Weinshall (2019) and Bengio et al. (2009) use a fixed scoring function and pace function for the entire training process. They empirically show that a curriculum helps to learn fast in the initial phase of the training process. In this section, we propose and analyze our novel DCL algorithm that updates the difficulty scores of all the examples in the training data at every epoch using their gradient information. We hypothesize the following: Given a weight initialization and a local minima obtained by full training of *vanilla* SGD, the curriculum ordering determined by our DCL variant leads to speedup in training. We

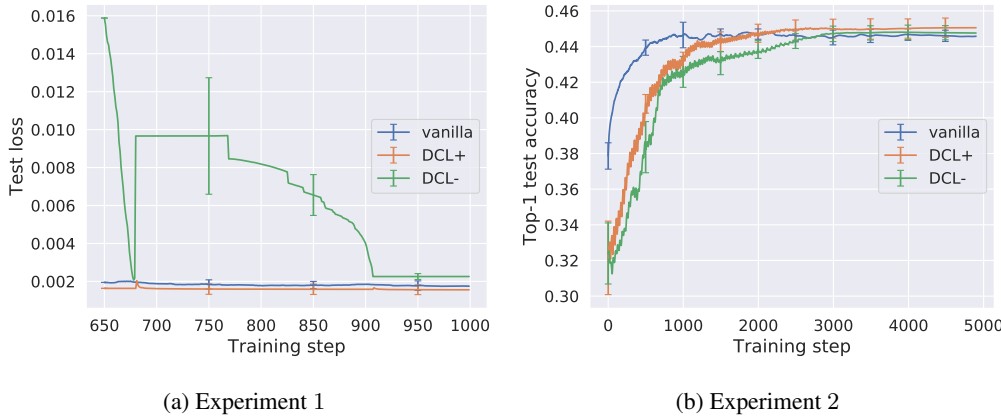

(a) Experiment 1            (b) Experiment 2

Figure 2: Learning curves of experiments 1 and 2 comparing *DCL+*, *DCL-*, and *vanilla* SGDs. Error bars signify the standard error of the mean (STE) after 30 independent trials.

first describe the algorithm, then the underlying intuition, and finally validate the hypothesis using experiments.

Our DCL algorithm iteratively works on reducing the L2 distance, $R_t$, between the weight parameter $\boldsymbol{w}_t$ and a given optimal weight $\bar{\boldsymbol{w}}$ at any training step $t$. Suppose, for any $\tilde{t} < t$, $S_{\tilde{t},t}$ is the ordered set containing the $(t-\tilde{t}+1)$ indices of training examples that are to be shown to the learner from the training steps $\tilde{t}$ through $t$. Let us define $\boldsymbol{a}_t = (\bar{\boldsymbol{w}} - \boldsymbol{w}_t)$, $R_t = \|\boldsymbol{a}_t\|_2$, and $\theta_i^{\tilde{t}}$ as the angle between $\nabla f_i(\boldsymbol{w}_t)$ and $\boldsymbol{a}_{\tilde{t}}$. Then, using a geometrical argument, (see Figure 1),

$$R_t^2 = \left( R_{\tilde{t}} - \eta \sum_{j=\tilde{t},\ i\in S_{\tilde{t},t-1}}^{j=t-1} \left( \|\nabla f_i(\boldsymbol{w}_j)\|_2 \cos\theta_i^{\tilde{t}} \right) \right)^2 + \eta^2 \left( \sum_{j=\tilde{t},\ i\in S_{\tilde{t},t-1}}^{j=t-1} \left( \|\nabla f_i(\boldsymbol{w}_j)\|_2 \sin\theta_i^{\tilde{t}} \right) \right)^2$$

$$= R_{\tilde{t}}^2 - 2\eta R_{\tilde{t}} \sum_{j=\tilde{t},\ i\in S_{\tilde{t},t-1}}^{j=t-1} \left( \|\nabla f_i(\boldsymbol{w}_j)\|_2 \cos\theta_i^{\tilde{t}} \right) + \eta^2 \left( \sum_{j=\tilde{t},\ i\in S_{\tilde{t},t-1}}^{j=t-1} \left( \|\nabla f_i(\boldsymbol{w}_j)\|_2 \cos\theta_i^{\tilde{t}} \right) \right)^2$$

$$+ \eta^2 \left( \sum_{j=\tilde{t},\ i\in S_{\tilde{t},t-1}}^{j=t-1} \left( \|\nabla f_i(\boldsymbol{w}_j)\|_2 \sin\theta_i^{\tilde{t}} \right) \right)^2 \tag{1}$$

For a *vanilla* model, $S_{0,T}$ is generated by uniformly sampling indices from $[N]$ with replacement. Since, finding a set $S_{0,T}$ to minimize $R_T^2$ and an optimal $\bar{\boldsymbol{w}}$ are intractable for nonconvex optimization problems, we approximate the DCL algorithm (*DCL+*, see Algorithm 1). We approximate $\bar{\boldsymbol{w}}$ with $\tilde{\boldsymbol{w}}$, which is a local minima obtained from training the *vanilla* SGD model. Also, to reduce computational expense while sampling examples, we neglect the terms with coefficient $\eta^2$ in equation 1 while designing our algorithm. Algorithm 1 uses a greedy approach to minimize $R_t^2$ by sampling examples at every epoch using the scoring function

$$score_t(\boldsymbol{x}_i) = -\|\nabla f_i(\boldsymbol{w}_t)\|_2 \cos\theta_i^t = -\frac{\boldsymbol{a}_t^{\mathrm{T}} \cdot \nabla f_i(\boldsymbol{w}_t)}{\|\boldsymbol{a}_t\|_2} = \rho_{t,i}. \tag{2}$$

Let us denote the models that use the natural ordering of mini-batches greedily generated by Algorithm 1 for training networks as *DCL+*. *DCL-* uses the same sequence of mini-batches that *DCL+* exposes to the network at any given epoch, but the order is reversed. We empirically show that *DCL+* achieves a faster and better convergence with various initializations of $\boldsymbol{w}_0$. We use learning rates with an exponential step-decay rate for the optimizers in all our experiments as traditionally

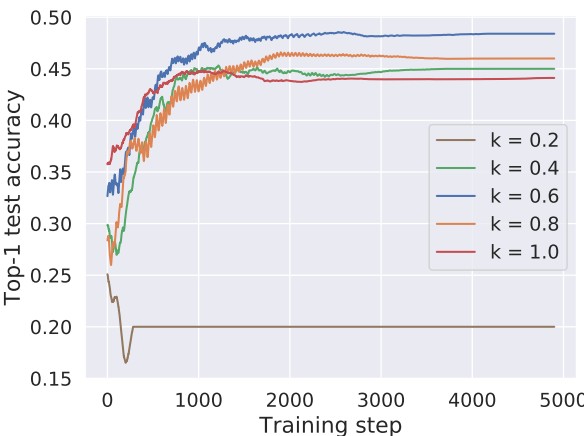

Figure 3: Learning curves for experiment 2 with varying $pace(t) = \lfloor kN \rfloor$ for *DCL+*. The parameter $k$ needs to be finely tuned for improving the generalization of the network. A low $k$ value exposes only examples with less noise to the network at every epoch whereas a high $k$ value exposes most of the dataset including highly noisy examples to the network. A moderate $k$ value shows less noisy examples along with some examples with moderate level of noise to the learner. Here, a moderate $k = 0.6$ generalizes the best.

done (Simonyan & Zisserman, 2014; Szegedy et al., 2016). For a fair comparison, we tune the learning rates and decay rates of the models.

**Experimental setup:** In our experiments, we set $pace(t) = \lfloor kN \rfloor \ \forall t$, where $k \in [b/N, 1]$ is a tunable hyper-parameter. We use a 2-layer fully-connected network (FCN) with 10 hidden neurons and Exponential Linear Unit (ELU) nonlinearities to empirically validate our algorithms ($k = 0.9$) on a subset of the MNIST dataset with class labels 0 and 1 (**Experiment 1**). Since, this is a very easy task (as the *vanilla* model accuracy is as high as $\sim 99.9\%$), we compare the test loss values across training steps in Figure 2a to see the behaviour of DCL on an easy task. *DCL+* shows the fastest convergence, although all the networks achieve the same test accuracy. *DCL+* achieves *vanilla*'s final test loss score at training step 682 ($\sim 30\%$ speedup). In **Experiment 2**, we use a 2-layered FCN with 128 hidden neurons and ELU nonlinearities to evaluate our DCL algorithms ($k = 0.6$) on a relatively difficult small_mammals dataset (Krizhevsky et al., 2009), a super-class of CIFAR-100. Figure 2b shows that *DCL+* achieves a faster and better convergence than *vanilla* with respect to the test set accuracy in experiment 2. *DCL+* achieves *vanilla*'s convergence test accuracy score at training step 1896 ($\sim 60\%$ speedup). Further experimental details are deferred to Appendix B.1.

Since, DCL is computationally expensive, we perform DCL experiments only on small datasets. Fine-tuning of $k$ is crucial for improving the performance of *DCL+* on the test set (see Figure 3). We fine-tune $k$ by trial-and-error over the test accuracy score.

## 4 WHY IS A CURRICULUM USEFUL?

At an intuitive level, we can say that *DCL+* converges faster than the *vanilla* SGD as we greedily sample those examples whose gradient steps are the most aligned towards an approximate optimal weight vector. In previous CL works, mini-batches are generated by uniformly sampling examples from a partition of the dataset which is made by putting a threshold on the difficulty scores of the examples. Notice that our *DCL* algorithms generate mini-batches with a natural ordering at every epoch. We design *DCL+* and *DCL-* to investigate an important question: can CL benefit from having a set of mini-batches with a specific order or is it just the subset of data that is exposed to the learner that matters? Figure 2 shows that the ordering of mini-batches matters while comparing *DCL+* and *DCL-*, which expose the same set of examples to the learner in any training epoch. Once the mini-batch sequence for an epoch is computed, *DCL-* provides mini-batches to the learner in the decreasing order of noise. This is the reason for *DCL-* to have high discontinuities in the test loss curve after every epoch in Figure 2a. With our empirical results, we argue that the **ordering of mini-batches within an epoch does matter**.

Bengio et al. (2009) illustrates that removing examples that are misclassified by a Bayes classifier ("noisy" examples) provides a good curriculum for training networks. SPL tries to remove examples that might be misclassified during a training step by avoiding examples with high loss. CL by transfer learning avoids examples that are noisy to an approximate optimal hypotheses in the initial phases of training. *DCL+* and *DCL-* try to avoid examples with noisy gradients that might slow

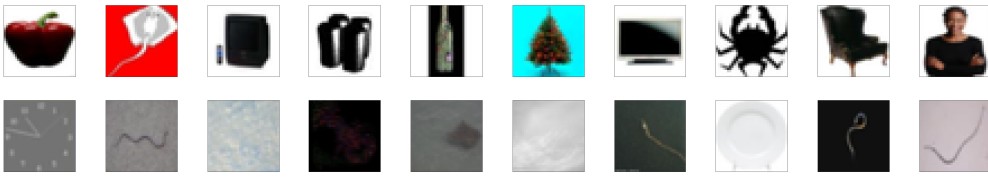

Figure 4: Top 10 images with the highest standard deviation values (top row) and top 10 images with the lowest standard deviation values (bottom row) in CIFAR-100 dataset.

---

**Algorithm 2** Curriculum learning method.

---

**Input:** Data $\mathcal{X}$, batch size $b$, scoring function $score$, and pacing function $pace$.
**Output:** Sequence of mini-batches $[B_0, B_1, ..., B_{T-1}]$.

1: sort $\mathcal{X}$ according to $score$, in ascending order
2: $B \leftarrow [\,]$
3: **for** $(i = 1;\ T;\ 1)$ **do**
4:      $size \leftarrow pace(i)$
5:      $\tilde{\mathcal{X}}_i \leftarrow \mathcal{X}[0, 1, ..., size - 1]$
6:      uniformly sample $B_i$ of size $b$ from $\tilde{\mathcal{X}}_i$
7: **end for**
8: **return** $B$

---

down the convergence towards the desired optimal minima. Guo et al. (2018) empirically shows that avoiding noisy examples improves the initial learning of convolutional neural networks (CNNs). According to their work, adding noisy examples to later phases of training serves as a regularizer and improves the generalization capability of CNNs. *DCL+* uses its pace function to avoid highly noisy examples (in terms of gradients). In our DCL experiments, the parameter $k$ is chosen such that few moderately noisy examples (examples present in the last few mini-batches within an epoch) are included in training along with lesser noisy examples to improve the network's generalization. We show the importance of tuning the pace function for *DCL+* in Figure 3. Hence, the parameter $k$ **serves as a regularizer and helps in improving the generalization of networks**.

## 5 STATISTICAL MEASURES FOR DEFINING CURRICULA

In this section, we discuss our simple approach of using task-specific statistical measures to define curricula for real image classification tasks. We perform experiments and validate our proposal over various image classification datatsets with different network architectures.

Based on the classification task, one could find a statistical measure that could serve the purpose of a scoring function for defining a curriculum. For instance, standard deviation and entropy are informative statistical measures for images and used widely in digital image processing (DIP) tasks (Kumar & Gupta, 2012; Arora, 1981). Mastriani & Giraldez (2016) uses standard deviation filters for effective edge-preserving smoothing of radar images. Natural images might have a higher standard deviation if they have a lot of edges and/or vibrant range of colors. Edges and colours are among the most important features that help in image classification at a higher level. Figure 4 shows 10 images that have the lowest and highest standard deviations in the CIFAR-100 dataset. Entropy measure gives a measure of image information content and is used for various DIP tasks such as automatic image annotation (Jeon & Manmatha, 2004). We experiment using the standard deviation measure ($stddev$), the Shanon's entropy measure ($entropy$) (Shannon, 1951), and different norm measures as scoring function for CL (see Algorithm 2). The performance improvement with norm measures is not consistent and significant over the experiments we perform (see Appendix A for details). For

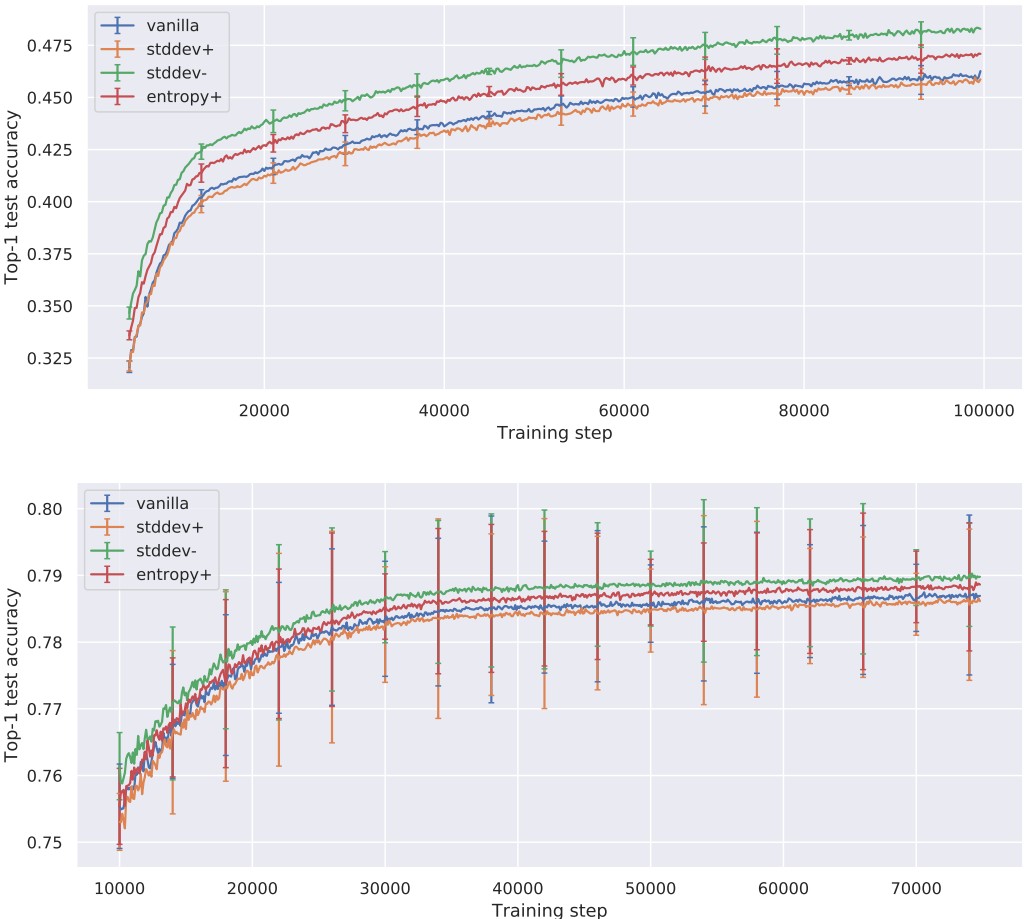

Figure 5: Learning curves for experiments 3 (top) and 4 (bottom). Error bars represent the STE after 25 independent trials.

a flattened image example represented as $\boldsymbol{x} = [x_{(0)}, x_{(1)}, ..., x_{(d-1)}]^{\mathrm{T}} \in \mathbb{R}^d$, we define

$$\mu(\boldsymbol{x}) = \frac{\sum_{i=0}^{d-1} x_{(i)}}{d} \qquad \text{and}$$

$$stddev(\boldsymbol{x}) = \sqrt{\frac{\sum_{i=0}^{d-1}(x_{(i)} - \mu(\boldsymbol{x}))^2}{d}}. \tag{3}$$

We use a fixed exponential pace function that exponentially increases the amount of data exposed to the network after every fixed $step\_length$ number of training steps. For a training step $i$, it is formally given as: $pace(i) = \lfloor \min(1, starting\_fraction \cdot inc^{\lfloor \frac{i}{step\_length} \rfloor}) \cdot N \rfloor$, where $starting\_fraction$ is the fraction of the data that is exposed to the model initially, $inc$ is the exponential factor by which the the pace function value increases after a step, and $N$ is the total number of examples in the data.

**Baseline and experimental setup:** We denote CL models with scoring function $stddev$ as $stddev+$, $-stddev$ as $stddev-$, $entropy$ as $entropy+$, and $-entropy$ as $entropy-$. Even though $vanilla$ is a competitive benchmark, we also use the CL by transfer learning framework (Hacohen & Weinshall, 2019) (denoted as $TL$) as a baseline. We employ two network architectures for our experiments: a) FCN-512 – A 2-layer FCN with 512 hidden neurons and ELU nonlinearities, and b) CNN-8 – A moderately deep CNN with 8 convolution layers and 2 fully-connected layers. We perform the following experiments: CNN-8 with a) CIFAR-100 (**Experiment 3**), b) CIFAR-10 (**Experiment 4**), c)

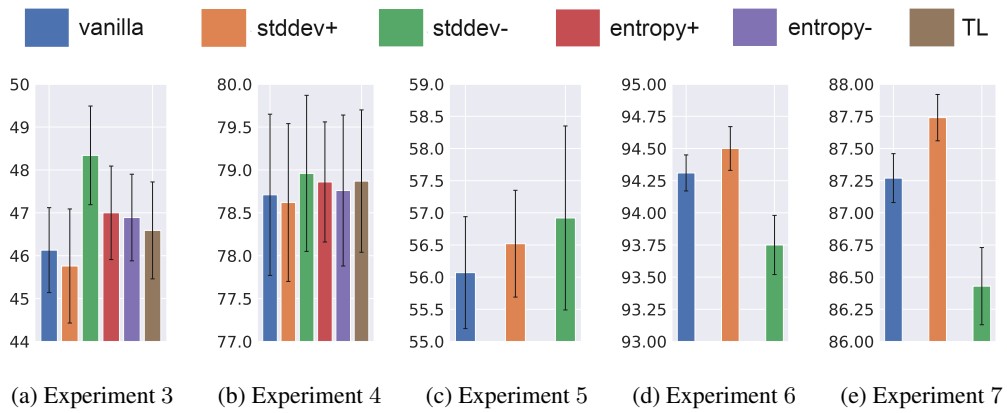

(a) Experiment 3  (b) Experiment 4  (c) Experiment 5  (d) Experiment 6  (e) Experiment 7

Figure 6: Bars represent the final mean top-1 test accuracy (in %) achieved by models in experiments $3 - 7$. Error bars represent the STE after $25$ independent trials for experiments $3, 4$ and $7$, and $10$ independent trials for experiments $5$ and $6$.

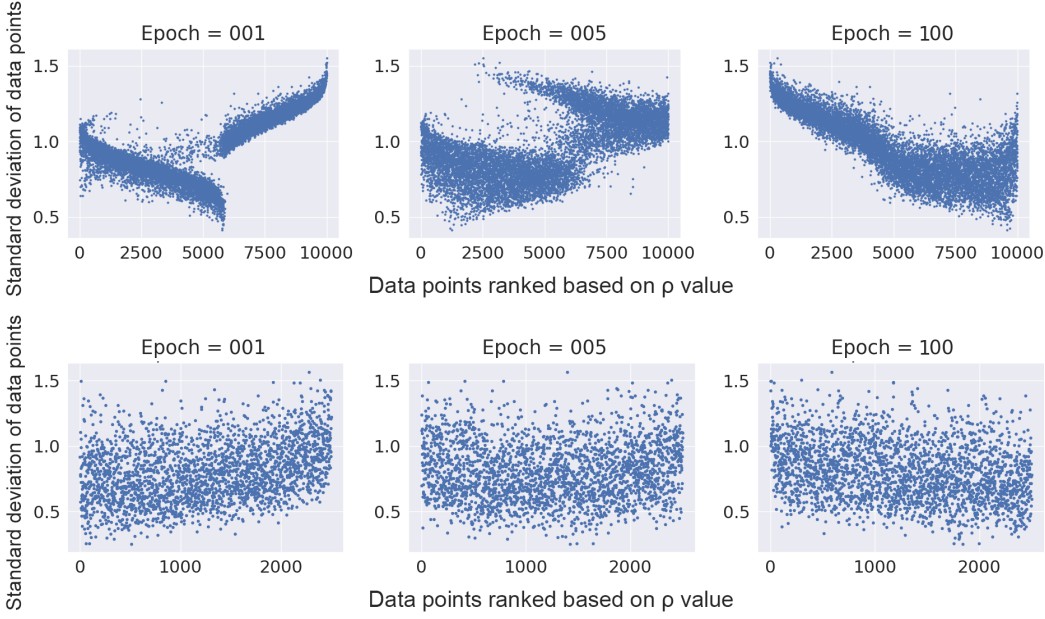

Figure 7: Relation of $\rho$ and $stddev$ values of examples over training epochs $1, 5,$ and $100$ for experiments $1$ (top row) and $2$ (bottom row).

small_mammals (**Experiment 5**) (these are the benchmark experiments used in Hacohen & Weinshall (2019)), and FCN-512 with d) MNIST (**Experiment 6**), e) Fashion-MNIST (**Experiment 7**). For experiments $3 - 5$, we use the same experimental setup as used in Hacohen & Weinshall (2019). More experimental details are deferred to Appendix B. In all our experiments, the models use fine-tuned hyper-parameters for the purpose of an unbiased comparison. Our experiments ($3$ and $4$) show that both $stddev$ and $entropy$ measures as scoring function provide superior results. Since $stddev$ performs the best, we further investigate its benefits on multiple tasks (experiments $5, 6,$ and $7$). Figures 5 and 6 show the results from our experiments. The best CL models achieve speedups of $56.6\%, 55.0\%, 46.9\%, 8.1\%,$ and $50.1\%$ on experiments $3 - 7$, respectively. With these empirical evidences we argue that $stddev$ is a good measure to define curricula for image classification tasks.

**Analyzing *stddev* with our DCL framework**: We use our DCL framework to understand why *stddev* works as a scoring function. We try to analyze the relation between the standard deviation

and $\rho_{t,i}$ values of examples over training epochs. Figure 7 shows the plots of standard deviations on the Y-axis against examples plotted on the X-axis ranked based on their $\rho_{t,i}$ values in an ascending order at various stages of training. It shows the dynamics of $\rho_{t,i}$ over initial, intermediate and final stages of training. Relation between $\rho_{t,i}$ and $stddev$ is evident from these plots. In the initial stage of training, examples with high standard deviations tend to have high $\rho$ values. In the final stage of training (the trend changes to the exact opposite after the intermediate stage), examples with high $\rho$ values tend to have low standard deviation values. This shows that $stddev$ can also be useful in removing noisy examples from the initial phases of training and hence help in defining a good curriculum.

## 6 CONCLUSION

In this paper, we propose two novel CL algorithms that show improvement in performance over multiple image classification tasks with CNNs and FCNs. Our DCL algorithm greedily samples data to move towards an optimal weight in a faster manner. It tries to avoid noisy gradients from slowing down the convergence. Its two variants, *DCL+* and *DCL-*, provide insights on how important ordering of mini-batches is for CL. The requirement to finely tune the pace function of *DCL+* shows that adding a moderate amount of noisy examples to training helps in improving the network's generalization capability. In this work, a fresh approach to define curricula for image classification tasks based on statistical measures is introduced. This technique makes it easy to score examples in a completely unsupervised manner without the aid of any teacher network. We thoroughly evaluate our new CL method and find it benefits from a faster (mean speedup of $\sim 43\%$) and better convergence (test accuracy improvement of $\sim 0.2\% - 2.2\%$). We use our DCL framework to understand $stddev$. With our results, we argue that CL algorithms help in faster initial learning by removing noisy examples that slow down the convergence towards a minima. Gradually, they add noisy examples to training in order to improve the performance of the network on unseen data.

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

## A  ADDITIONAL EMPIRICAL RESULTS

In Section 5, we study the performance of CL using $stddev$ and $entropy$ as scoring measures. Other important statistical measures are mode, median, and norm (Kumar & Gupta, 2012). A high standard deviation for a real image could mean that the image is having a lot of edges and a wide range of colors. A low entropy could mean that an image is less noisy. Norm of an image gives information about its brightness. Intuitively, norm is not a good measure for determining difficulty of images as low norm valued images are really dark and high norm valued images are really bright. We experiment with different norm measures and find that they do not serve as a good CL scoring measure since they have lesser improvement with higher accuracy variance over multiple trials when compared to $stddev$- on the CIFAR datasets. We use two norm measures

$$
\begin{aligned}
norm(\boldsymbol{x}) &= \|\boldsymbol{x}\|_2, \qquad \text{and} \\
class\_norm(\boldsymbol{x}) &= \|\boldsymbol{x} - \mu_{\boldsymbol{x}}\|_2
\end{aligned} \tag{4}
$$

where $\boldsymbol{x}$ is an image in the dataset represented as a vector, and $\mu_{\boldsymbol{x}}$ is the mean of all the images belonging to the class of $\boldsymbol{x}$. All the orderings are performed based on the scoring function and the examples are then arranged to avoid class imbalance within a mini-batch in our experiments. Let us denote the models that use the scoring functions $norm$ as *norm+*, $-norm$ as *norm-*, $class\_norm$ as *class_norm+*, and $-class\_norm$ as *-class_norm*.

Figure 8 shows the results of our experiments on CIFAR-100 and CIFAR-10 datasets with CNN-8 using $norm$ and $class\_norm$ scoring functions. We find that the improvements reported for *norm-*, the best model among the models that use norm measures, have a lower improvement than *stddev-*. Also, *norm-* has a higher STE when compared to both *vanilla* and *stddev-*. Hence, based on our results, we suggest that standard deviation is a more useful statistical measure than norm measures for defining curricula for image classification tasks.

## B  EXPERIMENTAL DETAILS

### B.1  NETWORK ARCHITECTURES

All FCNs (denoted as FCN-M) we use are 2-layered with a hidden layer consisting of M neurons with ELU nonlinearities. Experiment 1 employs FCN-10 while experiment 2 employs FCN-128 with no bias parameters. The outputs from the last layer is fed into a softmax layer. Experiments 6 and 7 employ FCN-512 with bias terms. The batch-size is 50.

For experiments $3 - 5$, we use the CNN architecture that is used in Hacohen & Weinshall (2019). The codes are available in their GitHub repository. The network (CNN-8) contains 8 convolution layers with $32, 32, 64, 64, 128, 128, 256,$ and $256$ filters respectively and ELU nonlinearities. Except for the last two layers with filter size $2 \times 2$, all other layers have a fliter size of $3 \times 3$. Batch normalization is performed after every convolution layer, and $2 \times 2$ max-pooling and $0.25$ dropout layers after every two convolution layers. The output from the CNN is flattened and fed into a fully-connected layer with $512$ neurons followed by a $0.5$ dropout layer. A softmax layer follows the fully-connected output layer that has a number of neurons same as the number of classes in the dataset. The batch-size is $100$. All the CNNs and FCNs are trained using SGD with cross-entropy loss. SGD uses an exponential step-decay learning rate. Our codes will be published on acceptance.

### B.2  HYPER-PARAMETER TUNING

For fair comparison of models, the hyper-parameters should be finely tuned as rightly mentioned in Hacohen & Weinshall (2019). We exploit hyper-parameter grid-search to tune the hyper-parameters of the models in our experiments. For *vanilla* models, grid-search is easier since they do not have a pace function. For CL models, we follow a coarse two-step tuning process as they have a lot of hyper-parameters. First we tune the optimizer hyper-parameters fixing other hyper-parameters. Then we fix the obtained optimizer parameters and tune other hyper-parameters.

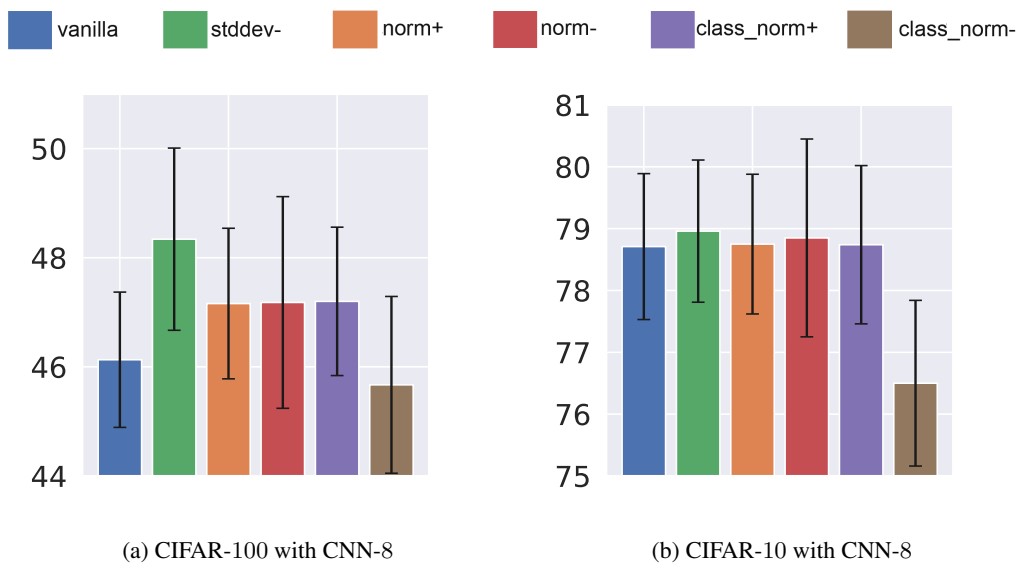

(a) CIFAR-100 with CNN-8    (b) CIFAR-10 with CNN-8

Figure 8: Bars represent the final mean top-1 test accuracy (in %) achieved by models. Error bars represent the STE after 10 independent trials.

### B.3 DATASET DETAILS

We use CIFAR-100, CIFAR-10, small_mammals, MNIST, and Fashion-MNIST datasets. CIFAR-100 and CIFAR-10 contain $50,000$ training and $10,000$ test images of shape $32 \times 32 \times 3$ belonging to 100 and 10 classes, respectively. small_mammals is a super-class of CIFAR-100 containing 5 classes. It has $2,500$ training and $500$ test images. MNIST and Fashion-MNIST contain $60,000$ training and $10,000$ test gray-scale images of shape $28 \times 28$ belonging to 10 different classes. All the datasets are pre-processed before training to have a zero mean and unit standard deviation across each channel.

