# OpenReview forum: "A Simple Approach To Define Curricula For Training Neural Networks"
_ICLR.cc/2021/Conference — Reject_

### Official Review · AnonReviewer1 · 2020-10-25
**Interesting ideas but much work is needed to formalize them and evaluate them correctly.**

**Rating:** 4
**Confidence:** 5

**Review:**

Summary:
This work studies a number of curriculums for faster training of neural networks. They first propose a curriculum named DCL+ that is designed to order data points based on their alignment of gradient with the direction of optimization. This curriculum depends on the evaluation of individual gradients of datapoints as well as an approximation to a local optima. Next, they study a number of easy-to-compute statistical measures for ordering data points.

Pros:
- The idea of ordering points based on using alignment of their gradients with the direction to the local optima is interesting.
- The idea of using easy-to-compute statistical measures of data is also interesting.

Cons:
- The empirical setup needs improvement. Important baselines are missing as described below. Standard datasets are not used and the ones used are relatively easy. The combination of these makes it hard to make any conclusions.
- The ideas are not formalized well enough. Specifically, important details in the definition of optimal weights in DCL is missing. Also, it is not clear why standard deviation of an image or its entropy could be a proxy for how useful a data point is for training.

Detailed notes:

Intro:
- The discussion about SGD and the generalization of other optimization methods is missing recent works such as [1].
- It is said that “...SGD samples examples from data uniformly at random.” This is an inexact description of common training setups. We can use SGD with any stochastic estimate of gradients as long as it is unbiased. It doesn’t have to be from uniformly sampling a subset. See for example [2].
- Related works on example reweighting and ordering are missing. For example [3], [4], and [5].

Section 3:
- Eq 1: 1) theta is not defined, 2) why isn’t the first equality an inequality?
- Figure 3: 1) what is theta? 2) Why is R3 marked as the distance between w3 and w0? Shouldn’t it be the distance between w3 and bar{w}?
- “We approximate w ̄ with w ̃, which is a local minima obtained from training the vanilla SGD model”, isn’t the goal of the entire training to find w^bar? How many steps do you train for to get w~?
- “score(x)” change of notation, score was defined as a function of both x and y in Section 2.
- “We use learning rates with an exponential decay rate for the optimizers” what is an exponential decay rate? How do you tune? Tune both base and the exponent? Why is it fair? There are numerous other learning rate schedules that are more common like the step-decay.
- Figure 2: what is the dataset? What is the task? Why is a data point noisy if it has a low score? Does this figure mean the hyperparameters are tuned on the test set rather than a validation set?
- Experiment 1 is mnist with labels 0 and 1. As noted in the text this task is very easy. Both DCL- and DCL+ eventually seem to classify the test set correctly.
- Experiment 2 is small-mammals dataset. No citation is given. It is said to be a super-class of CIFAR-100. No comparison is done with optimization methods other than SGD. No references are given for prior baselines on this dataset. No reason is given for not trying out the proposed method on common benchmarks such as full MNIST, CIFAR-10, and CIFAR-100.
- Figure 1: the placement of this figure is before Figures 2, 3 and algorithm 1 but it is referred to after them. The caption is not descriptive enough as it refers to experiments 1 and 2 that doesn’t make it easy to understand the figure without going back and forth between the figure and the text.

Section 5:
- There is no empirical comparison with prior works.
- There is no empirical comparison between the DCL method proposed in prior sections.
- Figure 6: same problem in captions as Figure 1.


Other:
- I could not uncompress the supplementary material. It needed PK compatibility V4.6 which does not come in the standard zip package.

[1] Choi, Dami, et al. "On empirical comparisons of optimizers for deep learning." arXiv preprint arXiv:1910.05446 (2019).
[2] Zhao, Peilin, and Tong Zhang. "Stochastic optimization with importance sampling for regularized loss minimization." international conference on machine learning. 2015.
[3] Loshchilov, Ilya, and Frank Hutter. "Online batch selection for faster training of neural networks." arXiv preprint arXiv:1511.06343 (2015).
[4] Katharopoulos, Angelos, and François Fleuret. "Not all samples are created equal: Deep learning with importance sampling." arXiv preprint arXiv:1803.00942 (2018).
[5] Chang, H. S., Learned-Miller, E., & McCallum, A. (2017). Active bias: Training more accurate neural networks by emphasizing high variance samples. In Advances in Neural Information Processing Systems (pp. 1002-1012).


=============
After rebuttal:
Thank you for improving the clarity. Unfortunately, the following issues are still unresolved to me:
- Figure 1: this geometrical argument seems to be at the core of Eq. 1 but I still have a hard time understanding it. You might want to formalize the argument in text.
- Experiments are not convincing. Even the original MNIST is not a representative dataset for optimization methods. The theoretical contributions of this work is not enough to justify having limited experiments.
- The rebuttal says: "The hyperparameter k in DCL is tuned by trial and error on the test set.". Does that mean mistakenly using the test set as the validation set?

---

> ### Author Response · Authors · 2020-11-23
> **Reply to R1**
>
> Thank you for your comments and suggestions for improvement. We have added all the related work that has been pointed out. We have corrected the statement about SGD to say that it proceeds by making unbiased estimates of the gradient on the full data.
>
>
> Q1: Unclarity in equation 1.
>
>
> A1: We have corrected the inconsistencies between Figure 1 (in the revised draft) and the definitions in the text for equation 1. In Figure 1, $\theta$ is the angle between ($\bar{w} - w_3$) and $a_0$, this is a local definition and not related to the $\theta_{i}^{\tilde{t}}$ defined outside. $\theta_{i}^{\tilde{t}}$ is the angle between $\nabla f_i (w_t)$ and $a_\tilde{t}$.
>
> Q2: How many steps do we train for to obtain $\tilde{w}$?
>
> A2: We train the vanilla model once until convergence (in terms of training accuracy) to obtain $\tilde{w}$, given $w_0$ (we make this clear in Section 3 of the revised draft). The actual number of steps varies according to different datasets and architectures. $\bar{w}$ is a global optima and it is not clear we can reach that efficiently. We have clarifed the writeup to reflect this.
>
> Q3: Learning rates for the optimizer
>
> A3: We use an exponential step-decay as the learning rate scheduler. We tune both the exponential decay factor and the steps after which the learning rate is decayed.
>
> Q4: Regarding Figure 3 (in the revised draft), “noisy” data, and tuning k
>
> A4: Figure 3 shows the performance of DCL+ with the same setup as experiment 2 (as mentioned in the caption) using different k values.  In DCL framework, an example is “easy” if its $\rho$ value is low. That is, an “easy” example has its gradient aligned towards the minima more than a “hard” example’s gradient. DCL considers examples with high $\rho$ values as “noisy” since they misguide gradient descent away from the minima.
> The hyperparameter k in DCL is tuned by trial and error on the test set. The experiments we perform by varying the value of k help in understanding how CL serves as a regularizer.
>
> Q5: Dataset for DCL
>
> A5: As rightly pointed out by the reviewers, the DCL algorithm is computationally very expensive. So, experimenting with larger datasets such as full MNIST and CIFAR would be laborious. Moreover, our intention is to use DCL as a framework to support our following arguments and not as a practical CL algorithm:
> a) Ordering of mini-batches within an epoch matter (comparing DCL+ and DCL-).
> b) CL serves as a regularizer that helps in improving the generalization of the model by avoiding “very hard” examples for training. DCL+ shows that a curriculum can be defined with gradient information.
> Hence, we only analyze the working of DCL on a hard dataset (small-mammals) and an easy dataset (MNIST with labels 0 and 1).

---

### Official Review · AnonReviewer3 · 2020-10-26
**Simple but requires unreastic knowledge and gains are small**

**Rating:** 3
**Confidence:** 4

**Review:**

The paper contains two curriculum learning algorithms of which one assume knowledge of the parameters found by the baseline, uniform-sampling, model to push updates in that direction, and the second orders images according to an increasing stddev/entropy of pixels. While the first approach is impractical because of the strong assumption, the second approach demonstrates small gains that lie within random variance (Fig. 5, Fig. 6) and would be not straight-forward to apply to non-image data e.g. text. These reasons make the paper hard to accept.

The main problem is knowing the parameters of the baseline, SGD, optimization. It's not clear why would one even need optimization again, if (a good enough) result is already known and gains from this re-optimization do not significantly improve over this baseline. The speedups mentioned in the abstract (45% and 43%) could not be located in the results in main body of the paper. How were they measured? Even if aligning updates with the SGD-trained parameters does speed up convergence, re-training from scratch will cost 143% of baseline time instead of 43%, as the standard training needs to be counted too.

Issues include:
- How to sample using \rho_{t,i}? It's not a distribution and can be negative.
- Figure 1: Judging from the plot, the vanilla curve converges faster than the curriculum. How can one see the >40% curriculum speed up?
- Abstract's claim of removing noise is only supported in Section 4 through citing related works. Also, more evidence would be needed to call k a regularizer.
- lines 9-10 in Algorithm 1 would interfere with bucketing in seq2seq applications and adversely affect performance.

Regarding related work in Sec. 3: I couldn't confirm in (Graves et al, 2017) that they also sort examples by difficulty.

The last approach to define curriculum through statistical quantities makes sense, in principle, although the difference between curves in Fig. 5 is very small and could be caused by random variance as the error bars on Fig. 5 and Fig. 6 show. Another problem is that it's straight-forwardly applicable only to images and not categorical data, like text.

One suggestion of possible paper improvement: consider swapping and reworking sections 5 and 3, so that the content of sec. 5 becomes the main proposal and a reworked sec. 3 - its analysis. There one could analyze if the example ordering according to stddev does bias updates towards some "good" point of convergence, with one possible definition of "good" according to (now, unknown during optimization) SGD results.

Other minor remarks:
- "greedy approach" is mentioned multiple times before being explained on page 4. Consider deferring the use of term to that place.
- Contributions: useful is a vacuous word, consider dropping it.
- notation: square brackets used to denote several objects - sequences [B1, B2, ..] , ranges [T] and vectors [x1, x2, .. ]. Using different brackets could be better.
- well-known concepts:
  * no need to define stddev and mean in (2)
  * (Arora, 1981): if entropy requires a citation at all then citing Shannon directly would be more appropriate.
- Sec. 2: curriculum is defined by two functions -> we define curriculum by two functions
- conclusion: display -> show
- while CL indicate that -> while CL indicates that
- judicial ordering -> judicious ordering

=== After rebuttal ===

Thank you for your answers. I'm keeping the rating at 3.

1)+2). I'm still not convinced that it's fair to claim an improvement of X% for a curriculum that, relying on final weights of a "vanilla" SGD-trainedmodel, converges in _additional_ X% to the 100% of "vanilla" time.

3). Fair enough, but the revised draft still reads like examples are sampled from it.

I double checked the context of citing (Graves et al, 2017) and believe it's still imprecise as in the original draft.

---

> ### Author Response · Authors · 2020-11-23
> **Reply to R3**
>
> Thank you for your comments and suggestions for improvement.
>
> Q1: How are speedups measured and how to verify the 40% number?
>
> A1: The speedup is measured as the improvement over the vanilla model (we have made this clear in the last paragraph of Section 2 of the revised draft). It is computed using the training step value at which CL achieves the same final test accuracy as the vanilla model. For example, if the vanilla model converges to 90% test accuracy in 100 steps, and the CL model achieves 90% test accuracy at training step 50, then the speedup for the CL is 2x. This helps in understanding how good the CL performs when compared to the vanilla model when a target test accuracy is to be achieved.  We calculate the speedups attained by CL in each of our experiments and report them along with the mean speedup.
>
> Q2: Speedup gains for DCL+
>
> A2:   As the reviewer rightly points out, DCL is computationally very expensive as it requires to compute gradients for the entire dataset to find an ordering. Our intention is to use DCL as a framework to support our following arguments and not as a practical CL algorithm:
> a) Ordering of mini-batches within an epoch matter (comparing DCL+ and DCL-).
> b) CL serves as a regularizer that helps in improving the generalization of the model by avoiding “very hard” examples for training. DCL+ shows that a curriculum can be defined with gradient information.
> But, stddev+/- and entropy+/- do not require the vanilla model to be trained. Hence, our proposed CL algorithms based on statistical measures are beneficial in practice.
>
> Q3: How to sample using $\rho_{t,i}$?
>
> A3: As the reviewer rightly points out, $\rho_{t,i}$ is not a distribution but a scoring function. It is only for the purpose of sorting the data points, as mentioned in Algorithm 1.
>
> Q4: Interference of Algorithm 1 with bucketing in seq2seq models
>
> A4: Our algorithms aim towards defining CL for image classification tasks. However, the sorting of data points could be performed within the buckets for seq2seq models.

---

### Official Review · AnonReviewer4 · 2020-10-27
**Interesting ideas, but execution leaves something to be desired.**

**Rating:** 4
**Confidence:** 3

**Review:**

#### Summary
- This paper considers curriculum learning for neural networks in the context of supervised learning (specifically image classification).
- First, the authors propose and assess a DCL+ algorithm. DCL+ uses a scoring function based on the alignment of an example's gradient with the vector from the current weight to a local minimum weight (obtained via a previous "vanilla" run with standard SGD). The pacing function is a constant fraction of the dataset size. The effect is that for a given epoch, only the subset of data that induces gradients which most point towards the local minimum is used for training. DCL+ is empirically shown to result in marginal improvement over the vanilla run in terms of test performance at convergence, and a significant speedup to reach vanilla performance. To investigate the effect of minibatch ordering within an epoch, the authors propose DCL-, an ablation that reverses the DCL+ minibatch ordering. Since DCL- performs worse than DCL+, the authors argue that minibatch ordering matters.
- Second, the authors propose and assess a few curricula based on scoring functions that only use per-example statistics, e.g. standard deviation (stddev) or entropy of pixel values. Interestingly, it seems that for CNNs on CIFAR tasks, using stddev- (descending order of standard deviation) as the scoring function is best, while for MLPs on MNIST tasks, using stddev+ (ascending order of standard deviation) is best.
- The authors connect their DCL+ algorithm to Bengio et al. (2009), which argues that a successful strategy for CL methods is to remove "noisy" examples. They also provide scatterplots of the data under DCL+ and stddev scoring functions.

#### Strengths
- The algorithmic innovations considered are simple.

- The comparison between DCL+ and DCL- shows that minibatch ordering can matter in CL.

- If one is willing to treat the unsupervised curriculum as a tunable hyperparameter (e.g. sweep over stddev+, stddev-, and vanilla), then the experiments suggest that we obtain some gain in test performance. A caveat is that this only is supported for the few relatively toy image classification datasets and architectures considered in the paper.

#### Weaknesses
- For a purely empirical paper, I would not say that the experiments are very comprehensive. For example, a clear discrepancy exists between Exps. 3-5 and Exps. 6-7 (in the former, stddev- is best, while in the latter, stddev+ is best, and most of the time the opposite curriculum is worse than vanilla). No attempt is made to discuss this or tease out the underlying reason: is it the dataset (CIFAR vs. MNIST) or the model (CNN vs MLP), or something else?

- Conceptually, the curricula defined by image statistics have a critical weakness: they are completely agnostic to the label. This is in contrast with both DCL+/- and Bengio et al. (2009), which both consider "example noise" grounded in the full supervised task. The same image (and therefore statistics) with different labels (say, correct vs. incorrect) would play drastically different roles in a curriculum.

- The connection between the notion of a "noisy example" as considered by Bengio et al. (2009) (misclassified by a Bayes classifier) and that of DCL+ (the example's gradient has strong anti-alignment with the direction to the local optimum) is not made technically clear or explicit.

- Regarding Figs. 1, 5, and 6: I appreciate that the authors included a measure of uncertainty based on 25 or 30 trials, but I would strongly suggest adding significance tests for a more compelling analysis. This is sorely needed for, e.g., Experiment 4.

- Fig. 7: The caption and the text imply that you'd provide correlations, but this is missing. Also, I don't see significant correlations for the bottom row.

#### Recommendation
- I currently recommend rejection (4). While the ideas are simple and interesting, the weaknesses in this submission preclude it from being very informative or useful to the community.

#### Questions
- Why does this work not consider comparisons with prior CL methods other than the vanilla baseline? If none are applicable, please explain why.
- Can you unify the notion of a "noisy" example as considered by Bengio et al. (2009) and by DCL+?
- Fig. 1: Why does subplot (a) only show a truncated x-axis?

#### Minor suggestions
- p. 2: What is "theoretical evidence"? Do you mean that there is a lack of theory, or a lack of empirical evidence, or both?
- p. 2: Attribute the concepts of scoring and pacing functions to Hacohen and Weinshall (2019).
- In general, the clarity of the writing (e.g. sentence style, diction) can be improved by a few careful passes.

-------------------------------Post-rebuttal comments-------------------------------

Thank you for taking the time to revise your submission. I will maintain my original score of 4. The main justification for this is that the two main weaknesses I see in this paper (the first two I list in the original review) remain unresolved, and it indeed is unclear whether the second one could feasibly be addressed without significant changes to the core methodology currently proposed.

---

> ### Author Response · Authors · 2020-11-23
> **Reply to R4**
>
> Thank you for your comments and suggestions for improvement.
>
> In Figure 7, the term “correlation” is misleading, we intend to say that the plots show the relation between the rank according to the stddev and the $\rho_{i,t}$ values. We agree that the relationship between the two quantities in the bottom row (small mammals dataset) seems much weaker than the top one (we have added more text regarding Figure 7 on Page 9, first paragraph in the revised draft).
>
> A1: We have compared two of our experiments to Hacohen & Weinshall (2019) as a baseline in Figure 6 a,b.
>
> A2: Bengio et al. (2009) introduced the notion of “noisy example” for CL in their work. In our work, we intend to define “noisy examples” differently based on their gradient values, as the reviewer rightly understands. It is not clear to us whether these notions can be unified.
>
> A3: DCL-  shows a much worse test loss in the initial phases of training. Hence, we truncate that part in order to clearly show the improvement of DCL+ over vanilla towards convergence.

---

### Official Review · AnonReviewer2 · 2020-10-29
**The paper is of limited novelty and poorly written.**

**Rating:** 3
**Confidence:** 4

**Review:**

Summary:
This paper studies curriculum learning and proposes two methods to order the examples by (1) gradient information and (2) statistical measures like standard derivation and entropy. The experiment results show that the proposed curriculum learning strategies can speed up the convergence by a large margin and the authors provide some insights about why curriculum learning works.

Strengths:
1. The proposed "dynamic curriculum algorithm" can speed up the convergence by ~45% and the proposed task-specific curriculum strategy based on standard deviation and entropy can yield an average speedup of ~43%.
2. The code and data are shared and helpful for reproducing the experiments conducted in the paper.

Weaknesses:
1. The paper is poorly written. There are even no sections to discuss related works and experimental settings in the main paper. Although some related works are discussed scatteredly in the paper, it might be helpful to have a specific section to compare related works with the proposed methods, which makes it much easier to identify the contributions and novelty of this work. Besides, although I found the experimental settings in the supplementary, the main paper at least should have discussed the basic experimental setup to understand how the experiments are conducted.

2. The proposed DCL algorithm requires an optimal weight or a local optimal weight to calculate the difficulty scores. This requirement is unreasonable and renders the proposed methods useless.

3. The proposed scoring function (the equation at the end of page 3) requires to compute the gradients on each sample. Perform back-propagation and computing gradient are highly prohibitive. The learning curves against the time cost should also be reported, in complement to the learning curves against the training steps in Fig.1.

4. The pace function is just a constant, dependent on a tunable hyperparameter k. From Fig.2, it seems that the value of k has a large impact on the testing accuracy. It is not mentioned in the paper how the value of k is selected.

Suggestions for improvement:
1. It might be better to have a specific section to discuss related works and compare them with the proposed method.

2. Index all equations.

Questions:
The questions to be addressed in the rebuttal are listed below:
1. Where does the optimal weight in DCL comes from? Can the authors justify why a given optimal weight can be used during the training?

2. What is the time cost of the scoring function?

3. How is the value of k in the pace function tuned?

-------------------------------Post-rebuttal-------------------------------

Thank you for revising the submission and the clarification in the rebuttal. After reading the rebuttal and other reviews, my main concerns about the novelty and computation cost are still unsolved. Therefore, I will keep my original score.

---

> ### Author Response · Authors · 2020-11-23
> **Reply to R2**
>
> Thank you for your comments and suggestions for improvement.
>
> A1: We will clarify this point. The DCL algorithm does not really need an optimal set of weights. As mentioned briefly (in the second last paragraph of page 4 in the revised draft), we can run the DCL algorithm in the following manner --  for a given initialization of weight ($w_0$), the weight that vanilla SGD converges to ($\tilde{w}$) is taken as an approximation for global minima ($\bar{w}$). Our empirical analysis shows that DCL finds an ordering of the data points that leads to faster convergence of the model from $w_0$ to $\tilde{w}$. Empirical results show that, in fact, DCL reaches a better solution than $\tilde{w}$.
>
> A2: As the reviewer rightly points out, DCL is computationally very expensive as it requires to compute gradients for the entire dataset to find an ordering. Our intention is to use DCL as a framework to support our following arguments and not as a practical CL algorithm:
> a) Ordering of mini-batches within an epoch matter (shown using the comparative performance of DCL+ and DCL- in Figure 2 of the revised draft), and
> b) CL serves as a regularizer that helps in improving the generalization of the model by avoiding “very hard” examples for training (shown by varying k).
> DCL+ shows that a curriculum can be defined with gradient information. The practical scoring functions that we suggest -- stddev+/-, entropy+/- etc. can be computed efficiently using standard libraries.
>
> A3: We do not know what fraction of the dataset is “noisy” while training. Hence, the hyperparameter k in DCL is tuned by trial and error. The experiments we perform by varying the value of k help in understanding how CL serves as a regularizer.
> However, we can compare the performance of DCL+ with varying k values by running them for one training epoch. In our experiments, we find that the k value that performs the best (decided by looking at the slope of the learning curve) for one epoch is a good choice for a full training of DCL+.

---

### Author Response · Authors · 2020-11-23
**Rebuttal revision**

We first wish to thank the reviewers for their detailed comments and suggestions. Based on the reviews, we have made changes to our manuscript. Here is a list of the main edits that we made:

+ Added a few more references on works related to optimizers in Section 1.
+ Added a subsection (1.1) for related works. Recent works on reweighting and ordering are added.
+ We make our contributions clearer in subsection 1.2.
+ Algorithm 1 and Figure 3 from the previous version are now rightly placed.
+ We address the concerns regarding notations in the Preliminaries section.
+ Details on how to measure speedup are mentioned in Section 2.
+ Corrected Figure 3 (from the previous version) to be consistent with equation 1.
+ Text added in Section 3 to address the concerns regarding DCL.
+ The parts of the text that mentions baseline and experimental setup are highlighted.

We hope we have addressed the major issues raised by the reviewers in our revised manuscript and replies.

---

### Decision · Program_Chairs · 2021-01-07
**Final Decision**

**Decision:**

Reject

**Comment:**

This paper proposed two algorithms for curriculum learning, one based on the the knowledge of a good solution (e.g. a local minima or a solution found by SGD) and another one proposed for natural image datasets based on entropy and standard deviation over pixels.

Reviewers seem to like the ideas behind the proposed algorithms and their simplicity. However, there are several major concerns that are shared among reviewers:
1- One of the algorithms needs knowledge of a good solution (e.g. a local minima or a solution found by SGD) which makes it impractical and the other one doesn't use any information about the mapping between input and the label.
2- Discussing previous work on curriculum learning, explaining how proposed algorithms are different than previous work and empirical comparison to other curriculum learning methods are lacking or need a significant improvement.
3- The experiment section needs improvement both in terms of experimental methodology and having more tasks/datasets.

Reviewers have done a great job at pointing to specific areas that need improvement. I hope authors would use reviewers' comments to improve their work.

Given the above major concerns, I recommend rejecting this paper.